# Optimizing Ki-67 and E-Cadherin Thresholds for Improved Grade and Stage Classification in Urothelial Bladder Cancer

**DOI:** 10.3390/jcm15010338

**Published:** 2026-01-02

**Authors:** Stefan Harsanyi, Zuzana Varchulova Novakova, Lucia Neuschlova, Stefan Galbavy, Lubos Danisovic, Stanislav Ziaran, Katarina Bevizova

**Affiliations:** 1Institute of Medical Biology, Genetics and Clinical Genetics, Faculty of Medicine, Comenius University in Bratislava, 811 08 Bratislava, Slovakia; stefan.harsanyi@fmed.uniba.sk (S.H.); zuzana.varchulova@fmed.uniba.sk (Z.V.N.); schwarzova27@uniba.sk (L.N.); lubos.danisovic@fmed.uniba.sk (L.D.); 2Institute of Forensic Medicine, Faculty of Medicine, Comenius University in Bratislava, 811 08 Bratislava, Slovakia; sgalbavy@ousa.sk; 3Department of Urology, Faculty of Medicine, Comenius University in Bratislava, 826 06 Bratislava, Slovakia; 4Institute of Anatomy, Faculty of Medicine, Comenius University in Bratislava, 811 08 Bratislava, Slovakia; katarina.bevizova@fmed.uniba.sk

**Keywords:** bladder cancer, p53, Ki-67, E-cadherin, cut-off analysis, ROC analysis, MIBC, tumor grade, biomarkers

## Abstract

**Background:** Bladder cancer exhibits substantial heterogeneity, and accurate discrimination between non-muscle-invasive (NMIBC) and muscle-invasive disease (MIBC), as well as between low-grade (LG) and high-grade (HG) tumors, remains essential for appropriate clinical management. Established immunohistochemical (IHC) markers, such as p53, Ki-67, and E-cadherin, could be used in a new setting, but standardized cut-off values and their combined predictive value remain unclear. This study aimed to identify optimal cut-offs for these markers and to evaluate whether biomarker combinations enhance the discrimination of tumor grade and stage. **Methods:** A retrospective dataset of 568 cases of bladder cancer was analyzed. For each case, the expression of p53, Ki-67, and E-cadherin was quantified, and tumors were classified as NMIBC or MIBC, and as LG or HG. ROC-based cut-off selection was performed using Youden’s J criterion with 10-fold stratified cross-validation. E-cadherin was modelled using an inverted scale to reflect biological loss. Logistic regression models were used to evaluate the discriminatory performance of single markers, two-marker combinations, and a three-marker model. Cross-validated AUC values and optimal thresholds were reported. **Results:** Ki-67 showed the strongest single-marker performance for predicting both MIBC (AUC 0.842) and HG disease (AUC 0.813), with optimal cut-offs of ≥40% and ≥30%, respectively. p53 demonstrated moderate discrimination (AUC 0.778 for MIBC and 0.776 for HG), while E-cadherin, evaluated on an inverted scale, showed acceptable performance (AUC 0.746 for MIBC; 0.780 for HG). Combining markers yielded modest improvements, with the best performance observed for Ki-67 + E-cadherin (AUCs of 0.851 for MIBC and 0.838 for HG). **Conclusions:** Ki-67 is the most effective single biomarker for distinguishing invasive and HG bladder cancer, while E-cadherin provides complementary value. A two-marker panel combining Ki-67 and E-cadherin, using appropriate cut-offs, offers the highest overall performance and may serve as a practical tool for enhanced pathological stratification.

## 1. Introduction

Bladder cancer is one of the most common malignancies of the urinary tract. More than 600,000 new cases are diagnosed, and over 220,000 die from the disease every year worldwide, with incidence rates still rising in several regions [1,2,3]. Europe remains one of the most affected areas, with an age-standardized rate of approximately 19 per 100,000 men and 4 per 100,000 women [2]. In the United States, BC accounts for approximately 83,000 cases annually and is the fourth most common malignancy in men [3]. Approximately 75% of cases present with non-muscle-invasive bladder cancer (NMIBC). In contrast, the remaining 25% present as muscle-invasive bladder cancer (MIBC), which is associated with higher morbidity, mortality, and economic burden.

Accurate stratification remains important because NMIBC and MIBC follow different clinical trajectories and require different management strategies [4]. Standard clinical factors, including stage, grade, tumor size, multiplicity, and recurrence history, provide helpful information but do not fully reflect the biological diversity of urothelial carcinoma. This has led to growing interest in analyzing established immunohistochemical (IHC) markers and using them to refine their diagnostic value. The most commonly studied IHC markers in BC include p53, Ki-67, and E-cadherin, each reflecting a different aspect of tumor biology [5]. However, their clinical use remains limited because no standardized cut-off values reliably distinguish meaningful subgroups of BC. Methodologies, staining procedures, and scoring systems-threshold definitions have varied, leading to variable conclusions.

Aberrant p53 expression is common in urothelial carcinoma and is linked to higher grade, advanced stage, increased progression risk, and worse survival [6]. Missense mutations usually cause nuclear p53 overexpression, while nonsense mutations or deletions lead to complete loss of staining. Both extremes have been associated with poorer outcomes, while low-level “wild-type” expression is usually seen in less aggressive tumors [5]. Ki-67 indicates proliferative activity, and higher values are associated with higher stage and grade, shorter recurrence-free intervals, and an increased risk of progression [7,8]. While several studies point to its prognostic value, published positivity thresholds vary considerably, from about 15% to over 50%, which limits comparability and routine use [8,9]. Loss of membranous E-cadherin expression is common in more invasive tumors and more frequent in MIBC than in NMIBC [10]. However, there is a lack of uniformity in the definition of reduced or lost expression between studies.

This study aims to determine the most effective and clinically meaningful cut-off values for p53, Ki-67, and E-cadherin to differentiate NMIBC from MIBC and to differentiate tumor grades within current urothelial carcinoma classification systems. By applying a unified methodology and statistically optimized thresholds, the study seeks to support a more reliable biomarker-based stratification approach for bladder cancer.

## 2. Materials and Methods

### 2.1. Study Design and Dataset

This retrospective observational study analyzed anonymized clinical and pathological data from patients with urothelial bladder cancer. The dataset included demographic variables (age, sex), tumor characteristics (tumor type, tumor count, stage, and grade), recurrence status, and IHC expression levels of p53, Ki-67, and E-cadherin. Grading was evaluated using a two-tier system (low grade vs. high grade), and pathological stage was categorized as NMIBC (Ta–T1) and MIBC (≥T2) [11]. Two independent pathologists did IHC analysis. In Figure 1 are representative microphotographs.

Tissue microarrays were prepared from formalin-fixed paraffin blocks, and two independent pathologists qualitatively confirmed the representativeness through side-by-side histopathological comparison of TMAs with the original sections of the whole tumor. Discrepancies were resolved by joint review to reach a consensus value, which was used for analysis.

Sections of 4 μm were prepared by microtome and then placed on poly-L-lysine-coated slides. Immunohistochemical analysis was carried out on prepared tissue samples using the Leica ST 5050 immunostainer (Leica Biosystems, Nussloch, Germany) with the avidin-biotin peroxidase method and diaminobenzidine as the chromogen, according to the manufacturer’s instructions. Microscopic assessment of the immunohistochemical staining for E-cadherin and Ki-67 was performed using primary antibodies. E-cadherin-NCH38 clone (DAKO, Glostrup, Denmark) and Ki-67-murine monoclonal clone MIB-1 (DAKO, Hamburg, Germany) with dilution performed using standard staining procedures. Expression levels for p53 and Ki-67 were recorded as the percentage of positive tumor nuclei in 5% increments. For E-cadherin, we assessed membranous staining and recorded the rate of tumor cells with intact circumferential membrane staining in 5% increments. Fragmentary/partial membrane or purely cytoplasmic staining was classified as negative for that cell. Intensity was not used in the primary analysis.

### 2.2. Clinical Endpoints and Data Preparation

All entries were checked for completeness and internal consistency. Staging categories were harmonized, and IHC markers were treated as continuous variables. No imputation was applied, and records with missing IHC data were excluded from analyses for the affected marker. Two clinically relevant binary endpoints were defined and used to evaluate the classification performance of p53, Ki-67, and E-cadherin:Stage: MIBC (≥T2) vs. NMIBC (Ta/T1);Grade: high grade (HG) vs. low grade (LG).

### 2.3. Cut-Off Selection

Each marker was tested for performance at multiple cut-off thresholds (range 0–100%). Later, for each marker and each endpoint, receiver operating characteristic (ROC) curves were generated, and the area under the ROC curve (AUC) with 95% confidence intervals was calculated. The Youden index was chosen a priori as the primary rule for cut-off selection. To reduce optimism bias, a 10-fold stratified cross-validation scheme was implemented. In each training fold, the optimal cut-off was determined using the Youden index and subsequently evaluated on the held-out fold. Final thresholds were reported as the median cut-off across folds, together with out-of-fold estimates of sensitivity, specificity, and AUC. P53 was analyzed first as a continuous, second as a pattern-aware model, with 0% counted as loss of expression, characteristic of HG tumors. Because reduced E-cadherin expression is biologically associated with a higher risk, ROC analyses for E-cadherin were performed using an inverted scale, so that lower expression corresponded to a higher predicted probability of the adverse outcome. Reported AUC values for E-cadherin are reported on this inverted scale.

### 2.4. Statistical Analysis

Data preparation and initial descriptive organization were performed using Microsoft Excel (Microsoft Corporation, Redmond, WA, USA). Statistical analyses, including ROC curve generation, logistic regression modeling, and hypothesis testing, were conducted using IBM SPSS Statistics for Windows, Version 29.0 (IBM Corp., Armonk, NY, USA). The analytical objective was to identify optimal cut-off values for each IHC marker to discriminate between NMIBC and MIBC, and HG and LG disease. Analyses were performed according to a prespecified workflow focused on ROC-based threshold selection and internal validation. The dataset is available in the Appendix A.

Descriptive statistics were used to summarize the cohort, employing medians and interquartile ranges for continuous variables and frequencies for categorical variables. Comparisons between NMIBC and MIBC groups were conducted using the Mann–Whitney U test for continuous variables and chi-square or Fisher’s exact tests for categorical variables. To assess whether each marker contributed independently to stage or grade, multivariable logistic regression models were fitted, with continuous marker values as predictors and age and sex as covariates. A secondary model incorporating the final binary cut-off was used to provide an interpretable effect estimate for the chosen threshold.

## 3. Results

A total of 568 patients with BC were included in the analysis. The median age was 69 years (range 32–98). The cohort was predominantly male, comprising 451 patients (79.4%). Most tumors were papillary (439 cases, 77.3%), with solid and mixed tumors accounting for the remainder.

The tumor stage distribution showed that NMIBC accounted for the majority of cases, with Ta accounting for 253 patients (44.5%) and T1 accounting for 101 patients (17.8%). For MIBC, T2 represented 202 cases (35.6%), while T3 and T4 accounted for 7 (1.23%) and 5 (0.88%) cases, respectively. HG tumors were more common, with 328 patients (57.7%) classified as HG, while 240 (42.3%) were LG. Tumor count was available for 472 patients, of whom solitary tumors accounted for 306 cases (64.8%), and the remainder had multifocal disease. Recurrence status indicated that 372 patients (65.5%) presented with primary disease, while 196 (34.5%) had a recurrent tumor.

IHC marker levels showed broad distributions, ranging from 0% to 100%. Median p53 expression was 30% (IQR, 15–60%). The median Ki-67 expression was 30% (IQR, 11–60%), and E-cadherin showed the opposite pattern, with a median expression of 75% (IQR, 40–80%).

### 3.1. ROC-Based Identification of Optimal Cut-Off Values for Tumor Stage

Figure 2 illustrates the ROC curves for p53, Ki-67, and E-cadherin in the identification of MIBC. Ki-67 had the highest discriminative performance (AUC 0.842), followed by p53 (AUC 0.778) and E-cadherin (AUC 0.745). The cross-validated optimal cut-offs are summarized in Table 1. A subgroup analysis for NMIBC to differentiate between Ta and T1 stages proved ineffective for Ki-67 and p53, with AUCs of 0.637 and 0.610, respectively. Results for E-cadherin were better with an AUC of 0.719 and a sensitivity/specificity ratio of 0.614/0.747.

### 3.2. ROC-Based Identification of Optimal Cut-Off Values for Tumor Grade

For differentiating high- vs. low-grade tumors, ROC curves (Figure 3) again demonstrated the strongest performance for Ki-67 (AUC 0.817), whereas p53 and E-cadherin showed comparable, moderate discrimination (AUC 0.776 and 0.783, respectively). The final cut-offs derived using Youden’s index are listed in Table 2.

### 3.3. Multivariable Analysis and Combining IHC Markers

In multivariable logistic regression, adjusting for age and sex, all three markers remained independently associated with both MIBC and HG disease when modeled as continuous variables (per 10% increase: p53 OR 1.15 [1.04–1.27]; Ki-67 OR 1.55 [1.38–1.74]; per 10-unit increase in E-cadherin OR 0.84 [0.77–0.92] for MIBC; and p53 OR 1.18 [1.06–1.32]; Ki-67 OR 1.41 [1.25–1.59]; E-cadherin OR 0.72 [0.64–0.80] for HG; all *p* ≤ 0.006). To assess incremental value, we evaluated marker combinations using 10-fold stratified cross-validation. Secondary models using the binary ROC-derived cut-offs confirmed their interpretability, with Ki-67 providing the strongest effect estimates for both endpoints.

For MIBC prediction, Ki-67 alone achieved a mean AUC of 0.842, which increased only modestly when combined with p53 (AUC 0.845) or E-cadherin (AUC 0.851), and in the complete three-marker model (AUC 0.851). For HG disease, Ki-67 alone yielded an AUC of 0.813, whereas combining it with E-cadherin improved the AUC to 0.838. Adding p53 did not produce a further appreciable gain (Table 3). These findings suggest that Ki-67 is the dominant discriminator in this cohort, with E-cadherin providing modest additional value, while p53 contributes only limited incremental information when combined with the other markers.

## 4. Discussion

In this study, we evaluated three routinely available immunohistochemical markers—p53, Ki-67, and E-cadherin—as tools for distinguishing NMIBC from MIBC and HG from LG. Using ROC analysis with cross-validation, we identified optimal cut-offs for each marker and examined whether marker combinations improved discrimination beyond single markers. Overall, our findings confirm the central role of Ki-67 and E-cadherin and suggest that p53 adds only limited incremental value in this context.

Ki-67 was the strongest single discriminator in our cohort, with cross-validated AUCs of 0.842 for MIBC and 0.817 for high-grade disease. These results are in line with several meta-analyses showing that high Ki-67 expression is associated with higher stage, higher grade, recurrence, and poorer survival in bladder cancer [7,8,9]. Ko et al. reported that Ki-67 overexpression in NMIBC was significantly associated with disease progression and worse recurrence-free survival [9]. At the same time, Tian et al. confirmed significant associations between Ki-67 and both adverse clinicopathologic features and outcome across 31 studies [7]. In BCG-treated NMIBC, He et al. also found that increased Ki-67 expression was associated with a higher risk of progression, though heterogeneity across studies was observed [8]. Our optimal cut-offs ≈ 40% for predicting muscle invasion and ≈30% for distinguishing high-grade tumors, are somewhat higher than the cut-offs commonly used in the literature, where thresholds in the 10–30% range are frequently reported [12]. This likely reflects differences in patient mix, technical protocols, and scoring. Importantly, the direction of association is consistent with prior work: higher Ki-67 indicates more aggressive disease. Warli et al. specifically examined Ki-67 as a predictor of muscle invasion and showed significantly higher Ki-67 labeling indices in MIBC compared with NMIBC, supporting our observation that proliferation strongly tracks with depth of invasion [12]. Together, these data and our results reinforce the view that Ki-67 is a robust marker of biological aggressiveness and a practical candidate for stratifying both stage and grade in clinical practice.

In our cohort, p53 had moderate discriminative ability (AUC 0.778 for MIBC and 0.776 for HG) and tended to be more specific than sensitive at the chosen cut-off (≈38%). This pattern mirrors the broader literature: p53 is biologically central and often associated with high-grade and advanced-stage, but its performance as a stand-alone prognostic marker has been inconsistent. A landmark meta-analysis by Malats and Real concluded that, despite extensive study, evidence remained insufficient to support routine use of p53 alterations as a robust prognostic marker in bladder cancer [6]. A systematic review by Tandon et al. similarly reported that p53 gene status is associated with poorer survival. Still, it emphasized substantial heterogeneity in methods and cut-offs, as well as the risk of publication bias [13]. At the IHC level, Chowdhury et al. showed that strong p53 expression was much more frequent in high-grade and higher-stage urothelial carcinomas than in low-grade tumors, with a higher frequency in the MIBC vs. NMIBC subgroup, consistent with our findings [14]. Our data, therefore, fit well with this pattern: p53 expression is clearly higher in aggressive tumors. It contributes to risk stratification, but when Ki-67 and E-cadherin are already included, it adds only modest incremental information to combined models.

E-cadherin, analyzed on an inverted scale (lower expression = higher risk), showed acceptable discrimination for both MIBC (AUC 0.745) and high-grade disease (AUC 0.783). This is consistent with meta-analytic evidence that reduced E-cadherin expression in bladder cancer is associated with worse overall survival, poorer recurrence-free survival, higher stage, and higher grade [15]. Xie et al. pooled data from more than 1500 patients and reported that reduced or lost E-cadherin expression was significantly associated with poor prognosis and advanced clinicopathologic features [15]. Earlier, del Muro et al. demonstrated that loss of E-cadherin expression, especially when combined with altered β-catenin, was an independent predictor of poor survival in multivariate analysis, and Serdar et al. found that abnormal E-cadherin staining correlated with disease recurrence, progression, and bladder-specific survival [16,17]. More recent work by Esmail et al. showed that E-cadherin expression was present in 65% of NMIBC cases but only 10% of MIBC cases, supporting its association with muscle invasion [18].

Our thresholds (≤67% for MIBC; ≤75% for high-grade disease) sit well within this biological narrative: partial loss of membranous staining appears sufficient to signal an EMT-like shift towards a more invasive phenotype. Alone, E-cadherin did not outperform Ki-67, but its independent associations with invasiveness and outcome across multiple studies explain why it adds complementary value when combined with proliferation markers [13,15,16,17]. We also asked whether combining markers offers a clinically meaningful gain over single-marker models. In our cohort, adding p53 to Ki-67 modestly increased AUC for predicting MIBC and high-grade disease, but the largest improvement was observed when Ki-67 was combined with E-cadherin. This two-marker model achieved AUCs of ≈0.85 for both endpoints, and adding p53 on top did not further improve performance. This is highly consistent with the study by Ziaran et al. [19], who examined p53, Ki-67, and E-cadherin in 224 patients and found that Ki-67 and E-cadherin were most strongly associated with cancer-specific survival, progression-free survival, and recurrence-free survival. In contrast, the additional contribution of p53 was more limited [19]. Similarly, Missaoui et al. reported that, in multivariable models including p53, p27, Ki-67, E-cadherin, and HER2, proliferation and adhesion markers were among the more informative predictors of aggressive behavior [20].

From a practical standpoint, these cut-offs are not intended to replace histopathologic staging and grading, but they may serve as adjunct “risk flags” in routine diagnostics, particularly in borderline cases or limited TURBT specimens where understaging is a concern. Specifically, Ki-67 ≥ 40% and/or reduced E-cadherin (≤67%) may increase suspicion of muscle-invasive biology even when a specimen is categorized as NMIBC, supporting careful correlation with cystoscopic findings, adequacy of sampling (including presence of muscularis propria), and multidisciplinary review. Likewise, Ki-67 ≥ 30% and/or E-cadherin ≤ 75% may support an HG interpretation in morphologically equivocal cases. Importantly, the intended clinical role is supportive rather than definitive. In NMIBC, Ta and T1 tumors showed statistically significant differences in Ki-67 and E-cadherin in our cohort, but single-marker discrimination between Ta and T1 remained modest, indicating that these markers should prompt closer pathological evaluation and appropriate follow-up planning rather than upstaging disease on IHC alone [21].

### Limitations and Future Directions

The study is limited by its retrospective nature and reliance on a single dataset. IHC scoring can vary across laboratories, and standardizing staining protocols and interpretation remains a challenge. Also, focal marker expression, particularly for heterogeneous markers, may be under- or over-represented in TMA cores. Because TMA-based sampling can shift absolute thresholds, these cut-offs should be considered calibration values. Future research should validate our proposed thresholds and two-marker panel in independent cohorts, ideally within multicenter prospective studies, and integrate these markers into broader models that include clinical variables and molecular subtypes.

## 5. Conclusions

In conclusion, among the three analyzed IHC markers, Ki-67 emerged as the strongest single discriminator for both NIBC and HG disease in bladder cancer. E-cadherin provided complementary information and, when combined with Ki-67, yielded the best-performing model (AUC of 0.851). p53, while biologically relevant, did not substantially enhance discrimination when combined. These findings suggest a streamlined two-marker panel (Ki-67 + E-cadherin) may offer a practical and effective approach for improving pathological stratification in bladder cancer. Prospective validation in independent cohorts is warranted before clinical implementation. 

## Figures and Tables

**Figure 1 jcm-15-00338-f001:**
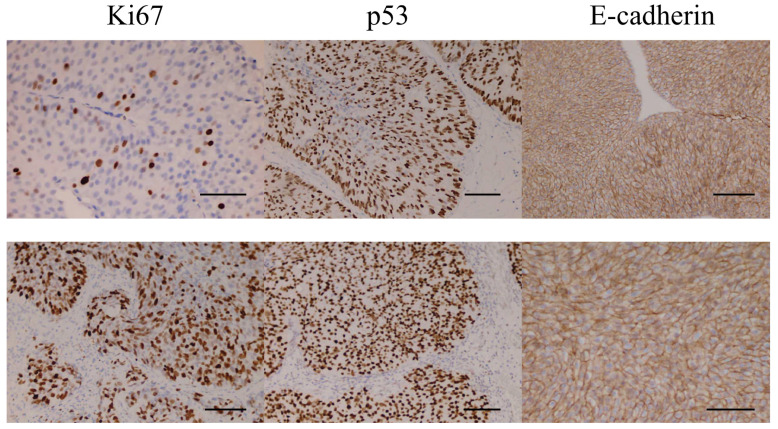
Microphotographs of the respective IHC staining. Upper = LG, non-invasive, lower = HG, invasive. Scale bars = 100 µm (are based on pixel calibration).

**Figure 2 jcm-15-00338-f002:**
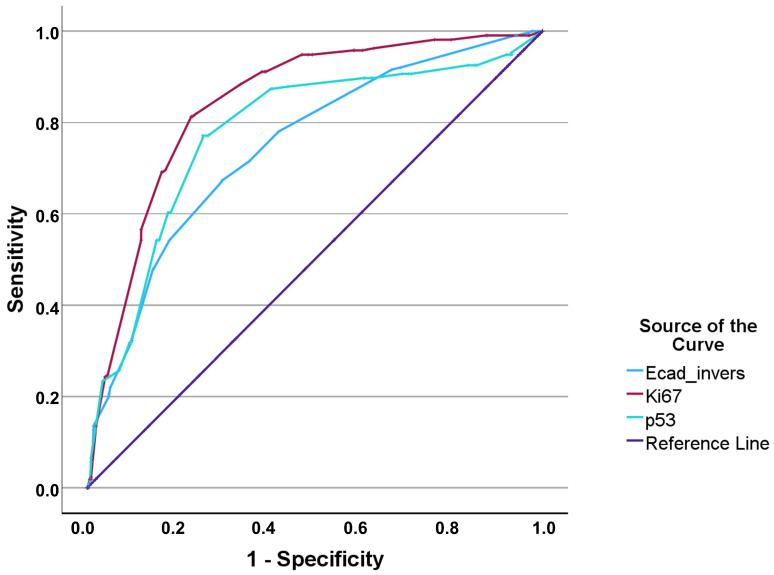
ROC curves for MIBC.

**Figure 3 jcm-15-00338-f003:**
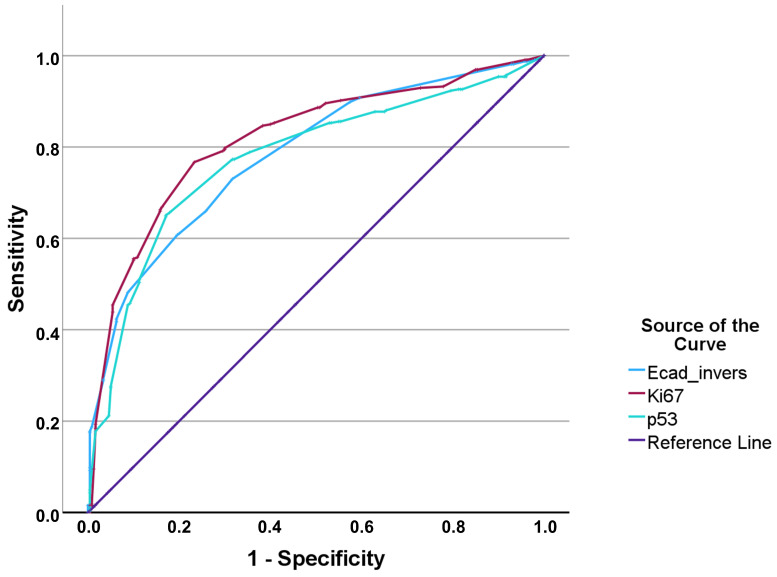
ROC curves for HG.

**Table 1 jcm-15-00338-t001:** Most effective cut-offs for NMIBC vs. MIBC.

Marker	AUC (95% CI)	Optimal Cut-Off	Direction	Sensitivity	Specificity
Ki-67	0.842 (0.807–0.873)	≥40%	Higher = higher risk	0.813	0.771
p53	0.778 (0.735–0.819)	≥38%	Higher = higher risk	0.774	0.746
E-cadherin	0.745 (0.704–0.785)	≤67%	Lower = higher risk	0.673	0.703

**Table 2 jcm-15-00338-t002:** Most effective cut-offs for LG vs. HG.

Marker	AUC (95% CI)	Optimal Cut-Off	Direction	Sensitivity	Specificity
Ki-67	0.817 (0.777–0.852)	≥30%	Higher = higher risk	0.768	0.767
p53	0.776 (0.738–0.814)	≥38%	Higher = higher risk	0.652	0.829
E-cadherin	0.783 (0.745–0.821)	≤75%	Lower = higher risk	0.732	0.683

**Table 3 jcm-15-00338-t003:** AUC for the combination of markers (10-fold CV, mean ± SD).

Marker Model		MIBC AUC (SD)	HG AUC (SD)
Single markers	p53	0.774 (0.094)	0.774 (0.043)
	Ki-67	0.842 (0.048)	0.813 (0.054)
	E-cadherin (low = risk)	0.746 (0.069)	0.780 (0.076)
Two-marker combinations	p53 + Ki-67	0.845 (0.057)	0.821 (0.050)
	p53 + E-cadherin	0.806 (0.068)	0.828 (0.053)
	Ki-67 + E-cadherin	0.851 (0.050)	0.838 (0.056)
Three-marker combination	p53 + Ki-67 + E-cadherin	0.851 (0.057)	0.842 (0.051)

## Data Availability

The minimal dataset is available in the Appendix A.

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
