# Peer review of "Optimizing Ki-67 and E-Cadherin Thresholds for Improved Grade and Stage Classification in Urothelial Bladder Cancer"

_jcm, 2026, doi:10.3390/jcm15010338_

Round 1

Reviewer 1 Report

Comments and Suggestions for Authors

The authors present a retrospective study evaluating the diagnostic performance of p53, Ki-67, and E-cadherin immunohistochemistry for discrimination between NMIBC and MIBC and between low- and high-grade urothelial carcinoma. Using ROC-based cut-off optimization with internal cross-validation, the study identifies Ki-67 as the strongest single marker and demonstrates modest but consistent improvement when combined with E-cadherin.   The manuscript is clearly written, logically organized, and addresses a clinically relevant problem. The cohort size is substantial, the statistical approach is appropriate, and the conclusions are generally supported by the data. With some clarification of methodological details and minor revisions.   Major issues   Methodology of Immunohistochemical Scoring The description of IHC evaluation would benefit from additional detail. While the authors state that expression was scored in 5% increments, it is not entirely clear whether scoring was based solely on the percentage of positive cells or whether staining intensity was also considered, particularly for E-cadherin where membranous completeness can be variable. In addition, two pathologists performed the assessments, but no measure of interobserver agreement is reported. Even a brief statement indicating concordance rates or explaining why agreement statistics were not calculated would strengthen the methodological rigor.   Use of Tissue Microarrays The study relies on tissue microarrays, which is practical for large cohorts but may not fully capture intratumoral heterogeneity, especially for proliferation markers such as Ki-67 and for E-cadherin expression. While representativeness is briefly mentioned, this issue should be discussed more explicitly as a limitation, as it may influence absolute cut-off values.   Statistical Reporting The ROC-based approach and use of cross-validation are strengths of the study. However, several results are described narratively without full statistical detail. In particular: Confidence intervals for AUCs are not consistently shown in tables. Multivariable logistic regression results are summarized without reporting odds ratios and confidence intervals. Providing these data, either in the main text or as supplementary material, would improve transparency and reproducibility. Clinical Interpretation The findings clearly support the dominant role of Ki-67 and the complementary value of E-cadherin. However, the manuscript would benefit from a more explicit discussion of how these cut-offs could be applied in daily diagnostic practice. For example, it would be useful to comment on whether the proposed thresholds might assist in borderline cases or influence surveillance or treatment

Comments on the Quality of English Language

Minor grammatical issues (e.g., spacing, hyphenation such as “cutoff” vs “cut-off”)

Author Response

The authors present a retrospective study evaluating the diagnostic performance of p53, Ki-67, and E-cadherin immunohistochemistry for discrimination between NMIBC and MIBC and between low- and high-grade urothelial carcinoma. Using ROC-based cut-off optimization with internal cross-validation, the study identifies Ki-67 as the strongest single marker and demonstrates modest but consistent improvement when combined with E-cadherin. The manuscript is clearly written, logically organized, and addresses a clinically relevant problem. The cohort size is substantial, the statistical approach is appropriate, and the conclusions are generally supported by the data. With some clarification of methodological details and minor revisions.  

  • We thank the reviewer for on-point comments, we tried to address them as well as we could.

Major issues  

Methodology of Immunohistochemical Scoring

The description of IHC evaluation would benefit from additional detail. While the authors state that expression was scored in 5% increments, it is not entirely clear whether scoring was based solely on the percentage of positive cells or whether staining intensity was also considered, particularly for E-cadherin where membranous completeness can be variable.

  • now better explained in materials and methods. sadly, we did not measure intensity

In addition, two pathologists performed the assessments, but no measure of interobserver agreement is reported. Even a brief statement indicating concordance rates or explaining why agreement statistics were not calculated would strengthen the methodological rigor.

  • It was a qualitative confirmation, and discrepancies were reviewed for consensus

Use of Tissue Microarrays

The study relies on tissue microarrays, which is practical for large cohorts but may not fully capture intratumoral heterogeneity, especially for proliferation markers such as Ki-67 and for E-cadherin expression. While representativeness is briefly mentioned, this issue should be discussed more explicitly as a limitation, as it may influence absolute cut-off values.  

  • We agree. Our study used one TMA core per tumor, which is practical for large cohorts but can under-sample spatial heterogeneity. We have added this as a limitation of our study.

Statistical Reporting

The ROC-based approach and use of cross-validation are strengths of the study. However, several results are described narratively without full statistical detail.

In particular: Confidence intervals for AUCs are not consistently shown in tables.

  • now inserted and adapted AUCs to have 3 decimals

Multivariable logistic regression results are summarized without reporting odds ratios and confidence intervals. Providing these data, either in the main text or as supplementary material, would improve transparency and reproducibility.

  • now included

Clinical Interpretation

The findings clearly support the dominant role of Ki-67 and the complementary value of E-cadherin. However, the manuscript would benefit from a more explicit discussion of how these cut-offs could be applied in daily diagnostic practice.

For example, it would be useful to comment on whether the proposed thresholds might assist in borderline cases or influence surveillance or treatment

  • explicit discussion of practical use is now included in the discussion

Reviewer 2 Report

Comments and Suggestions for Authors

This manuscript presents a retrospective analysis of 568 bladder cancer patients aimed at optimizing immunohistochemical (IHC) cut-offs for p53, Ki-67, and E-cadherin. The study addresses a relevant clinical need: the lack of standardized threshold values for these common biomarkers. The statistical approach, utilizing Youden’s J index and 5-fold stratified cross-validation, is a notable strength of the study. However, there are significant concerns regarding the interpretation of p53 expression patterns and the validation of Tissue Microarrays (TMAs) that should be addressed prior to publication.

The study utilized TMAs prepared from paraffin blocks. Given that bladder cancer—particularly high-grade disease—can be notoriously heterogeneous, reliance on TMAs presents a risk of sampling bias. While the authors state that "representativeness was confirmed by histopathological comparison... with the original sections", no quantitative data is provided to support this assertion. Please include metrics (e.g., concordance rates) to validate this comparison.

Regarding p53, did the authors observe a "null" pattern (0% staining) in High-Grade (HG) or Muscle-Invasive (MIBC) cases? The manuscript treats markers as continuous variables, implying a linear relationship where higher expression equals higher risk. However, since nonsense mutations can lead to complete loss of staining (0%), a linear model may incorrectly classify high-risk "null" tumors as low-risk. The authors should discuss how the "null" phenotype affects the ROC analysis or clarify if 0% staining was handled distinctively.

It would be highly valuable to see a sub-analysis (or at least a discussion) regarding the markers' ability to distinguish Ta from T1 tumors. The current analysis groups both stages as NMIBC. If Ki-67 is a marker of invasion, is there a significant difference in mean Ki-67 expression between the Ta (n=253) and T1 (n=101) patients in this cohort?

The scale bar is missing in the microphotographs. Please include scale bars and specify the original magnification in the figure legend.

Author Response

This manuscript presents a retrospective analysis of 568 bladder cancer patients aimed at optimizing immunohistochemical (IHC) cut-offs for p53, Ki-67, and E-cadherin. The study addresses a relevant clinical need: the lack of standardized threshold values for these common biomarkers. The statistical approach, utilizing Youden’s J index and 5-fold stratified cross-validation, is a notable strength of the study. However, there are significant concerns regarding the interpretation of p53 expression patterns and the validation of Tissue Microarrays (TMAs) that should be addressed prior to publication.

  • We thank the reviewer for the most constructive comments; we tried to answer all to the best of our ability.

The study utilized TMAs prepared from paraffin blocks. Given that bladder cancer—particularly high-grade disease—can be notoriously heterogeneous, reliance on TMAs presents a risk of sampling bias.

  • We agree with the reviewer that intratumoral heterogeneity, particularly in high-grade urothelial carcinoma, represents a limitation of TMA-based analyses. The use of TMAs was chosen to enable standardized, parallel immunohistochemical evaluation across a large retrospective cohort under identical staining conditions.

While the authors state that "representativeness was confirmed by histopathological comparison... with the original sections", no quantitative data is provided to support this assertion. Please include metrics (e.g., concordance rates) to validate this comparison.

  • The representativeness was assessed qualitatively by two independent pathologists through side-by-side histopathological comparison of TMA cores with corresponding whole-tumor sections, focusing on tumor grade, architecture, and cellular composition. No formal quantitative concordance metrics were calculated.

Regarding p53, did the authors observe a "null" pattern (0% staining) in High-Grade (HG) or Muscle-Invasive (MIBC) cases? The manuscript treats markers as continuous variables, implying a linear relationship where higher expression equals higher risk. However, since nonsense mutations can lead to complete loss of staining (0%), a linear model may incorrectly classify high-risk "null" tumors as low-risk. The authors should discuss how the "null" phenotype affects the ROC analysis or clarify if 0% staining was handled distinctively.

  • yes there were 34 null cases (14HG / 20LG ….. 10 MIBC / 24 NMIBC) we did use pattern-aware analysis for each marker, virtually any cut-off that we could find published before. However, continuous analysis gave better results than the “0% and above 50% model” – now stated in the cut-off section of materials and methods

Continuous p53- AUC for MIBC vs NMIBC: 0.778.   AUC for HG vs LG: 0.776.

Pattern-aware- AUC for MIBC vs NMIBC: 0.702.   AUC for HG vs LG: 0.675.

It would be highly valuable to see a sub-analysis (or at least a discussion) regarding the markers' ability to distinguish Ta from T1 tumors. The current analysis groups both stages as NMIBC. If Ki-67 is a marker of invasion, is there a significant difference in mean Ki-67 expression between the Ta (n=253) and T1 (n=101) patients in this cohort?

  • we now include this shortly in 3.1 section; but the analysis doesn’t prove any effective differentiation between Ta and T1, AUC only 0.64

The scale bar is missing in the microphotographs. Please include scale bars and specify the original magnification in the figure legend.

  • scale bars are added in